# Technology Support for Collaborative Preparation of Emergency Plans [note 1]

**DOI:** 10.3390/s19225040

**Published:** 2019-11-19

**Authors:** Nelson Baloian, Jonathan Frez, Jose A. Pino, Sergio Peñafiel, Gustavo Zurita, Alvaro Abarca

**Affiliations:** 1Department of Computer Science, Universidad de Chile, Santiago 837-0456, Chile; jpino@dcc.uchile.cl (J.A.P.); elxsergio@gmail.com (S.P.); 2School of Informatics and Telecommunication, Universidad Diego Portales, Santiago 837-0190, Chile; jonathan.frez@mail.udp.cl; 3Management Control and Information Systems Department, Faculty of Economics and Business, Universidad de Chile, Diagonal Paraguay 257, Santiago 833-0015, Chile; gzurita@fen.uchile.cl; 4Faculty of Engineering, Universidad Andrés Bello, Antonio Varas 810, Santiago, Chile; l.abarcacollao@uandresbello.edu

**Keywords:** collaborative decision support, geographical information systems, emergency planning, Dempster-Shafer theory

## Abstract

Preparing a plan for reaction to a grave emergency is a significant first stage in disaster management. A group of experts can do such preparation. Best results are obtained with group members having diverse backgrounds and access to different relevant data. The output of this stage should be a plan as comprehensive as possible, taking into account various perspectives. The group can organize itself as a collaborative decision-making team with a process cycle involving modeling the process, defining the objectives of the decision outcome, gathering data, generating options and evaluating them according to the defined objectives. The meeting participants may have their own evidences concerning people’s location at the beginning of the emergency and assumptions about people’s reactions once it occurs. Geographical information is typically crucial for the plan, because the plan must be based on the location of the safe areas, the distances to move people, the connecting roads or other evacuation links, the ease of movement of the rescue personnel, and other geography-based considerations. The paper deals with this scenario and it introduces a computer tool intended to support the experts to prepare the plan by incorporating the various viewpoints and data. The group participants should be able to generate, visualize and compare the outcomes of their contributions. The proposal is complemented with an example of use: it is a real case simulation in the event of a tsunami following an earthquake at a certain urban location.

## 1. Introduction

Specialists agree that a disaster management process consists of four phases: preparedness, mitigation, response and recovery [1]. Preparedness is the phase where specialists develop plans for preparing the population and the environment to react as resiliently as possible against disasters before they occur. A typical example for this phase would be to determine beforehand which could be the evacuation routes for the population in case of a tsunami in a coastal region. In order to develop plans for procedures and actions that should be followed in the event of an emergency often requires exploring, analyzing and evaluating georeferenced data, which means geography related data becomes of fundamental importance. Consequently, Geographical Information Systems (GIS) frequently support the preparation phase. When developing an action plan in order to react to an emergency due to a large-scale fire, an earthquake, a tsunami or flood experts often (if not always) use a GIS system for supporting the work, since they must always consider the features of the corresponding terrain and all possible entrance and exit paths. These terrain features should allow rescue teams to access the threatened place for helping people and/or bringing people to safer locations. This analysis should also take into consideration several factors requiring various knowledge domains, such as knowing the location of medical facilities, schools and safe open areas, the capacities of transportation in terms of vehicle flux and weight, the required times to move people from a certain location to another one, etc. [2]. Given these characteristics, we may describe the process of preparing a plan for rescue actions in the case of certain types of emergencies as a spatial decision-making scenario.

Decision Support Systems (DSS) are defined as interactive computer-based systems that help decision makers in the use of data and models to solve unstructured problems. A simplified model for the decision making (DM) process includes the following stages: (1) identifying the problem, (2) identifying and modeling the objective(s) of the decision, (3) collecting, generating and/or combining data to generate alternative scenarios, (4) evaluating options according to the objectives, (5) choosing an option and performing a sensitivity analysis. If the decision makers assess there is enough information, the process ends with a final decision; otherwise, the flow goes back to steps 2 or 3 [3]. Like Artificial Intelligence, the criteria for deciding which topics belong to DSS seem to be diffuse. Nevertheless, most authors trying to define DSS agree that a crucial feature is that human judgment is a key factor in the decision-making process cycle, generating options, re-defining and re-modeling objectives. This is a task involving creativity, which cannot be mechanized. Computers in turn, can provide humans with gathered data, generation of various decision options, and support for evaluating their outcome according to the goals. Machines can also provide visualization and communication of the results to other humans.

Frequently, DSS must deal with ill-structured problems; thus, the goals might not be very clear and/or there is insufficient information to solve them in a certain optimal way. Furthermore, ill-structured problems typically involve many stakeholders and require several decision makers; therefore, the solution tends to be subjective and unique. In addition, DSS systems include various modeling and analysis techniques to be used by non-computer experts. Hence, a DSS must be interactive, flexible and adaptable in order to support various solution approaches. Moreover, DSSs oriented to spatial problems must take into account that spatial information is inherently fuzzy and uncertain [4], which implies that fuzzy analysis techniques are needed.

There are many decision problems involving spatial issues that might be solved with the help of a GIS. The recurrent problem can be generally stated in the following way: Find a suitable area to “do” something. For example, Ghayoumian et al. [5] explain how to find specific locations for constructing artificial water recharge aquifers using floods. In this case, decision makers must not only be experts in aquifer recharge, but they will also need historical information and spatial data in order to design a formula, which reflects the correct criteria for selecting the suitable area(s). This formula is used to build a suitability map using a GIS. This map typically shows the suitability level on each point of the map satisfying the requirements. However, in ill-structured problems this criterion is complex to build because the goals are not clear and the various decision makers will tend to define different goals according to their own knowledge or expertise.

Another difficulty with GIS-based ill-structured problems is that the data available to the decision makers is frequently unreliable. In particular, data may be incomplete (not covering the whole space) and/or uncertain (there are doubts on the data accuracy and veracity). This situation is known as epistemic uncertainty [6]. One example concerns the evacuation of people from coastal areas after a strong earthquake; there is a high probability of a tsunami and decision makers must decide the evacuation procedure. Various options may be available, but the data to make an easy decision may not be at hand: how many people will be located in each portion of the territory at a certain time of the day? Will they have operational means of transportation at each location? Will they have basic supplies (water, electricity, gas…) at each location? etc.

Preparation for an emergency case (that is, making plans beforehand to face the situation in the event an emergency occurs) has been typically described in the literature as a collaborative decision making process since many experts should be involved [7,8,9]. However, there is no previous work mentioned in the literature aimed exclusively at supporting a group of experts doing this work. Although there are many works developed for supporting collaborative decision-making (see [10] for an example), emergency preparedness has some particular aspects, which justify the development of a dedicated tool: geographical context, simulation, discussion and ill-structured problem statement. The goal of this work is to fill that gap. 

The authors of this work have presented a preliminary approach showing a way to deal with incomplete and uncertain spatial data in a previous work [11]. That work introduced the theory for merging the criteria of various experts assessing the risk of a certain geographical area adapting the Dempster-Shafer theory for a geographic context and using mathematical operations, previously introduced by several different authors in the past. Then in [12] we presented a work in progress about a possible design of a prototype. In a later work [13], we discussed ideas regarding how to synchronize the work of various experts who are dealing with the problem of analyzing a scenario for emergency preparedness. In this work, we present an actual implementation of the ideas presented in the previous two papers and we further present a more refined manner in which experts can actually present in a very flexible and versatile way various scenarios of what can happen in an emergency. The presented system allows each expert to develop her own scenario(s) individually, discuss them asynchronously with the other experts and then combine their findings in order to present a single unified solution.

The rest of the paper is organized as follows: Section 2 reviews previous work from various perspectives. Section 3 deals with Dempster-Shafer theory and its application to spatial decision making. Section 4 presents a case study of collaborative decision making in emergency preparedness. Section 5 discusses combination methods. Section 6 introduces a collaborative decision-making tool for supporting discussions on preparedness for emergencies. Finally, Section 7 presents the conclusions.

## 2. Background

### 2.1. Scenario Analysis 

Scenario analysis is the process of evaluating possible future events and their consequences through the consideration of possible alternative states of the world (scenarios). These alternative states may not be equally likely. The definition used by the Intergovernmental Panel on Climate Change (IPCC) is a representation of scenarios applied to the natural sciences [14]: “A scenario is a coherent, internally consistent and plausible description of a possible future state of the world. It is not a forecast; rather, each scenario is one alternative image of how the future can unfold”.

Heugens and van Oosdterhout [15] define scenarios as “Stories about the future”. Instead of trying to predict the future, scenarios are possible descriptions of what the future might look like. Scenario development (or “analysis” or scenario planning) is a systematic method to creatively think about dynamic, complex and uncertain futures, and identify strategies to prepare for a range of possible outcomes [16,17]. The scenarios could focus on identifying “favorable futures” in which people wish to work or “unfavorable futures” that people might want to avoid or at least be prepared to face them. Instead of trying to reduce uncertainty through increasingly accurate predictions, scenarios can be a flexible way to discover potential surprises and prepare plans for an uncertain future that is in the essence of any complex system [18,19].

The scenario-planning concept originated during World War II, as an initiative of the US Air Force planners in an effort to predict their opponents’ actions; it allowed them prepare alternative plans for use in a particular scenario [20]. This planning can serve now various functions, e.g., helping in investigations, facilitating public learning and discussion, and helping to make political decisions. For each situation, the degree of stakeholders’ participation can be very varied. The scenarios are particularly useful in systems that are highly complex and unpredictable and/or where it is not possible to experiment by manipulating the system to see how the situation changes in response to certain changes in the values of the relevant parameters describing the state of the scenario [16].

Unlike sensitivity analysis, the objective of scenario-development is to produce a small number of scenarios with possible descriptions of system factors that can potentially be vastly different in each scenario. Sensitivity analysis tends to produce a high number of simulations resulting from a number of gradual variations of a single variable.

Categorizing scenarios can also help experts recognize them, capture their nature and reuse generalizations that they may derive from possible solutions. Categorizing also promotes communication during collaborative work between stakeholders, making the design of activities more accessible to the wide variety of experts that can contribute to this analysis [20]. In addition, they promote the background and plot to handle exercises for emergencies. The first step when designing a scenario is to determine the type of threat, danger, or situation that is happening [21]. According to [22], the scenarios may be classified as follows:*Exploration Scenarios*. Two types of exploration scenarios are commonly used in scenario planning: Projection and Possible Futures scenarios. In the Projection Scenarios, projects progress over time according to trends experienced in past periods, while in the Possible Futures Scenarios expert try to anticipate the next significant changes of variables.*Anticipation Scenarios*. These scenarios are based on different favorable or unfavorable visions of the future that could be reachable or avoidable. These scenarios may be proposed by expert judgment, where researchers and experts propose models of future conditions, or where stakeholders define the assumptions about the future that should be included in the scenarios.

The existing literature shows us that scenario analysis is a very pertinent process to conduct for emergency preparedness. Scenario analysis is also frequently collaborative. 

### 2.2. Maps and GIS-Based Decision Systems

According to [23], visualizing information may support a decision making process by using adequate visual representation of the information in order to detect patterns. In several areas, practitioners and professionals have historically used maps for supporting problem solving and decision-making processes through the visual representation of the geographical space [24]. In our days, maps are considered as artifacts that facilitate complex human activities involving the use, access and organization of geospatial information, [25]. Geo-visualization is a discipline that emerges from “Geographic Information Science” (GIScience), which goes beyond simple graphic representations of geo-spatial information, also involving the integration of knowledge construction with geo-spatial information and the design of user interfaces [25].

In this context, a relevant problem to study is the strategy to merge the work done by various participants in a group. A simple solution is to divide the work into parts each assigned to a person and then combining the results at a meeting or by a designated person. Another trivial solution is to take turns to consecutively improve an initial draft. Unfortunately, these easy solutions may not be applicable to all situations.

Baeza-Yates and Pino [26] presented a case in which the simple solutions are unfeasible. The problem concerns collaborative information retrieval: several people seek data and then each result is merged using a specific strategy to generate the result. Smeaton et al. [27], proposed a second solution to the same problem: various persons share a single user interface and cooperatively state queries and analyze results. Thirdly, Pickens et al. [28] suggested a solution, in which an algorithmically mediated collaborative search engine coordinates user activities during the searching session. Notice none of these solutions is trivial and each of them was developed in an ad-hoc way to the specific problem.

GIS-based decision-making also needs convergence research. A frequently occurring setting consists of a chauffeured group of decision makers, i.e., only one person can provide input to a GIS system perhaps through just one keyboard and mouse but with common output through a large shared display. This asymmetric situation has been criticized because the stakeholders are not equally able to contribute to the final decision (e.g., [29]). Moreover, for several public/private geographically related problems, various researchers have pointed out the need to democratize the decision-making process by involving the general public who are directly affected by the decisions (e.g., [29,30,31]). As a consequence, GIS-based systems have been often used to support collaborative decision-making processes. 

A decision-making process supported by GIS typically starts with two inputs: data and expert knowledge. Models are built using an expert’s knowledge, and alternative scenarios are constructed using different data input. These scenarios can be compared because they are based on the same model, although each model is based on different expert knowledge. Nevertheless, the knowledge can change during the evaluation process, for instance by including or removing a person from the experts’ team. A change in the knowledge of the experts’ team is challenging to represent in the model because it leads to changes in the used databases [32]. Spatial Decision Support systems are especially relevant to a vast number of scientific, economic and humanistic areas. The most common areas found in the literature are:Socio-economics: urban planning, industrial planning, agricultural land use, housing, education, natural resources, and many new smart city applications.Environmental: forestry, fire and epidemic control, floods and earthquake predictions, pollution, and smart city applications.Management: Organization logistics, electricity and telecommunication network planning, real-time vehicle tracking, and other public services planning like health services, security, fire protection, and smart city applications.

According to Malafant and Fordham [33], a GIS is always a DSS because it is used to support some stage of a decision-making process. According to [4], GIS offers appropriate techniques for data management, information extraction, routine manipulation, and visualization. However, Geographical Information Systems do not have the necessary analytical capabilities to manage a decision-making process. Furthermore, in [34] authors claim that at the time of publication (2010), the existing Spatial DSS tools (i.e., Decision Support Systems for spatial related problems) do not provide the needed characteristics, and recent literature does not show progress in this issue. This is exactly the direction in which the work presented in this paper contributes to this area. 

GIS-based systems play an important role when developing action plans in case of an emergency [29]. Very often, a suitability map displays the necessary information. This map shows the “appropriateness” of a certain terrain to fulfill certain conditions in a graphical way. For example, the risk of a landslide occurrence can be represented on a map by painting the areas with high risk with red color, medium risk with yellow and low risk with white. In order to build a suitability map, it is necessary to assign a certain value to each geographical point, which will determine its color. In the case when the suitability value is determined by the knowledge of experts, the Dempster-Shafer’s Plausibility Theory [35], has proven to be an effective tool [11,36,37,38]. 

### 2.3. Emergency Preparedness

In this sub-section we review the literature about emergency preparedness in order to find out which are the key aspects of this process and identify the research gaps that still exit. In particular, we want to investigate whether there have been computational systems developed to support this process. For this purpose, we made a web search on academic databases using the keywords “emergency preparedness”. From the obtained results we selected the most recent ones (from year 2015 onwards), except for three works ([39,40,41], which are from the years 2008, 2008 and 2003 respectively), since they are very much related to our work. Table 1 summarizes the works we found in this review.

Examining Table 1 we can conclude that research on emergency preparedness has focused mainly in proposing processes for performing this task, especially in recent times. We can also observe that more than half of the articles consider that collaboration among various actors is necessary. However, we found no proposals to develop a dedicated system to support this task, which is the focus of this work. By contrast, we can find many works in the literature presenting systems (many of them designed for mobile devices) for supporting responsive actions during an emergency occurrence or shortly afterwards (mitigation) [55].

## 3. Dempster-Shafer Theory and Spatial Decision Making

In the DSS area, there are mathematical tools designed to include expert knowledge and manage incomplete and uncertain information. Over the last 20 years, there have been studies using Belief Functions in different spatial decision-making problems [5,38] getting good results. However, the calculation method and results in these experiments are based on a specific spatial problem. As a result, functions used to evaluate the suitability of a place differ from each other and thus are hard to replicate or adapt to other spatial problems.

The Dempster-Shafer Theory (DST) [35] is a generalization of the Bayesian theory of probabilities that includes uncertainty as a primary element for the process. Let X to be the set of all possible outcomes of a process; the theory defines a mass assignment function (or simply a mass) as a function that assigns a value between 0 to 1 to each subset of X, satisfying that the sum of the masses of all subset is equal to 1. The theory defines two metrics that can be used to measure the support for an outcome and they are associated with the probability. The belief is defined as the total evidence to support an outcome, and the plausibility is defined as the total amount of evidence that can support an outcome. They define the lower and upper bounds for the probability of the outcome.

Multiple evidence sources expressed by their mass assignment functions can be combined using the Dempster Rule [56]. This rule establishes how to combine two masses m1 and m2 resulting in a new mass assignment function m3 which consider the evidence and uncertainty of both sources. The combination rule also defines a coefficient K, which is called the conflict and measures the level of disagreement these sources have about the process. One remarkable aspect of the combination rule is that it allows combining as many evidence sources as necessary to obtain the most accurate result.

Unlike probability, DST can express complex scenarios due to the inclusion of uncertainty in their computations. DST also allows models to express that they cannot make a prediction accurately. For example, consider a classification problem between classes A and B; classical probability models are forced to choose one of these classes to predict. In the same problem, DST can choose among A, B and the uncertainty (complete set), so if the model cannot predict an observation, then the uncertainty will be high.

In DST, hypotheses are statements that affect the final value of the target variable by reinforcing or weakening its occurrence, and they are associated with a probability of being correct. Then for each hypothesis and using the available geographic data, the model computes three values associated with the given probability:Plausibility: is the probability that the random variable takes values within the range of the query.Certainty: is the probability that the whole range of the distribution of variable is within the range of the query.Uncertainty: no valuable information can be derived from this data.

By defining these hypotheses, along with the mass assignment functions, experts “codify” their knowledge into the expert system based on Dempster-Shafer theory. For example, if the expert is looking for people density as the variable to predict, one hypothesis can be “people are in shops with a 20% of belief” or “people are in schools or workplaces with a 40% of belief”. We also defined query hypotheses: “people are in shops just like in place X, Y”. In the hypothesis statement, the expert can define multiple hypotheses, which are combined using Dempster-Shafer combination rules. Furthermore, the expert can design these complex scenarios without requiring any kind of GIS expertise.

DST has been adapted to geographical contexts by Frez et al. [57]. This work presents a framework for predicting the value of a variable that depends on time and space (geographically) using DST. In summary, the proposed model uses geographic data and a set of hypotheses or rules as the input. Using the hypotheses and the available data, the model builds several mass assignment functions for the prediction of the variable in a geographical region. In the same work, authors propose a variation of the Dempster Rule that considers the influence a set of geographic places where hypotheses apply may have on another geographical place for which the mass is being computed. The final mass for this place considers a combination of all influences taking into account the distance between them. The result of this process is a suitability map showing the predictions for the variable for a certain period.

The result of combining the hypotheses with real data and fuzzy techniques for spatial representation is a suitability map. A suitability map typically shows the suitability level on each point of the map that satisfies the requirements; in our case, it shows the belief degree of the hypothesis for each evaluated location. This kind of suitability map is what we call a simple scenario.

To illustrate this process, the region of interest is usually discretized into a grid. To see the effect of a hypothesis, consider one that states that “people are in stores with 40% of belief”; then we can query the geographical data to check where the stores are located, e.g., stores could be situated in the blue dots on the Figure 1a. After this, the belief is computed using the combination rule for each grid; the result of this computation is shown on Figure 1b.

In the example above, only one hypothesis was tested and computed. As we explained before, many hypotheses can be combined in order to obtain the final suitability map. The integration is performed by comparing the predicted values of each cell in the grid with the corresponding ones in the result of the other hypotheses; the combination considers the values both hypotheses predicted and the weight they have according to the statement of the hypothesis. Figure 2 illustrates the process of combining two belief maps obtained from the evaluation of a hypothesis. When comparing two cells, there are two options, both sources agree in their result and then the combination keeps their value, or the sources differ in results and then they have conflict, which implies uncertainty increases, and the value of the variable decreases.

## 4. Case Study

Since Chile is a country where earthquakes are very frequent, and a significant part of its territory is coastal land, plans for evacuating people in case of tsunamis are very important. During the 2010 earthquake, which had magnitude 8.8, over 500 casualties occurred because people did not have time to seek refuge in safe areas [58]. Therefore, it is very important to develop plans and train the population to evacuate the seaside and seek for safety in higher grounds. While developing such plans, many factors should be considered, like where people could be at the time of an earthquake and the speed at which a tsunami reaches the coast.

The preparation of an effective population evacuation plan requires the collaboration of various experts and stakeholders. Each one may have different opinions and hypotheses about which are the best options to elaborate an evacuation plan. We will assume there are five experts available in order to exemplify the proposed collaboration. Suppose two of them believe the best evacuation method is for people to go to higher grounds, using any possible routes and means of transport. The remaining three experts have another hypothesis: most people will not be able to reach the higher grounds before the tsunami arrives, so they should seek refuge inside high buildings (vertical evacuation).

Organizations responsible for dealing with emergencies typically rely just on traditional GIS to evaluate the aforementioned options. These systems provide information about the population living in the area, the number of schools, and other stored information.

The tsunami evacuation problem is a difficult one. In fact, it can be classified as an ill-formulated problem, because there is no information about the number of people who must be evacuated at the time of earthquake occurrence. There is no knowledge either on the precise available time from the earthquake occurrence to the subsequent tsunami reaching the coast. Of course, the population in the area may vary according to the time of the day, and day of the week. Furthermore, we must assume an earthquake can occur at any time.

Using the Dempster-Shafer Theory, we can build a set of hypotheses, which can tell us where people may be located. For example, let us consider a concrete case. There is an area of the coastline of Iquique (northern Chile) where there is a high belief that people will congregate there in large numbers during daytime, because this area includes universities, restaurants, shopping centers, a popular beach, etc. This area is also far from higher grounds. The vertical evacuation hypothesis may be appropriate for this area, as stated by three of the experts mentioned above. However, an obvious problem will occur if there are too many people and not enough high buildings.

The incomplete information and multiple scenarios make Dempster-Shafer theory a suitable choice to be applied to this case. We propose to encapsulate it in a Collaborative Geographical Information System (CGIS). The above-mentioned five experts could use this CGIS to make their own hypotheses evaluation. Because of this process, they will have a set of suitability maps. For example, the first two experts may disagree on where they believe people may be at different hours of the day. The other three experts may also disagree on how tall the buildings must be or the kinds of construction that must resist the tsunami being hypothesized.

There will be at least five suitability maps after each expert builds his/her own simple scenario(s). The next natural step will group the suitability maps according to both basic evacuation strategies. However, this procedure should be complemented by a collaborative step so that the experts desirably generate a single suitability map incorporating all contributions. Another situation that can occur is that all individual suitability maps are based on similar hypotheses, e.g., that during daytime people are in commercial areas, schools, universities, libraries, banks, bus stations, etc., but one expert may believe there will be more people at commercial areas, while others may say there will be more people located in residential areas.

The collaborative step can be designed to allow the combination of the various suitability maps generated by experts in a hierarchical order, according to certain operators, as Merigó and Casanovas [59] have suggested. Of course, the suitability maps to be combined using certain operation would need to be collaboratively decided by the experts. In each combination step, the resulting suitability map should be the outcome of the discussion of each scenario possibility including the experts’ hypotheses concerning known information about the area, and relevant factors on which to focus. 

Suitability maps are combined as a result of argumentation and discussion (Figure 3).

## 5. Combination Methods

A complex scenario is the combination of various simple scenarios. Building a complex scenario requires cooperative work between different stakeholders like experts in the particular scenario area and decision makers. In order to provide useful tools for collaborative scenario building for a single area we must divide the work in two different dimensions: Hypothesis Dimension and Time Dimension. The hypothesis dimension directly relates to the collaboration process between stakeholders (decision maker and experts) who have different hypotheses about belief function values for a certain time, e.g., one expert will have a certain hypothesis about the number of people at commercial areas for the morning, noon, evening and night. When combining suitability maps, experts should consider the same time dimension for stating their hypotheses. As the stakeholders discuss, argue, evaluate and combine the maps, the time dimension of the suitability map generation unfolds. 

In this work, additional to using the Dempster-Shafer combination rule, we propose to provide the user with four other operators, one of them with four variants, to collaboratively build a scenario, based on combinations of suitability maps which were previously constructed, either individually or already as a result of a collaborative process. The need to use these new operations appears from the fact that they have a semantic meaning, which can be more understandable for users who are not expert in the plausibility theory. In this way, they will feel keener to “merge” their findings. In general, the operations work in the following way: two or more input scenario maps of the same region, with belief values already assigned for each cell are merged in order to produce a single output scenario map. In the output map, the belief value for each cell results from applying the operation to the value of the same cell of the input maps. The operations are the following: *Sum*: The sum is probably the simplest operator a decision team should be able to use; it consists of summing the belief value of each scenario for each evaluated location (cell in the scenario). Graphically, it consists of summing the mass values of the same cell. Visually the resulting map does not show the sum of the two bars, one over the other, because the final values for each cell are normalized. This operator can be useful when three independently but related scenarios must be merged. For example, criminality, transit and street maintenance scenarios must be combined to evaluate the governmental resources needed in a general and comparative scope. Using sum the decision maker can easily identify the need of resources for each location independently of the type of need.*Subtraction*: The subtraction operator subtracts the belief value of two or more scenarios at each evaluated location. This operator can be useful when it is necessary to evaluate the differences between one scenario and others. For example, if we have a possible flood scenario and a refuge scenario. Using subtract, the decision maker can easily identify the places of refuge with lower flood belief values. *Average*: The average operator is the simple average between the belief degrees of each location in each scenario. The result of this operator is visually similar to sum but can be numerically different. For example, if a cell has value 0 for the belief for two of the experts’ maps and 100 for another, the sum will be 100, but the average will be 33.3. This operator can be used in order to find places in which to deploy scarce resources. One example is that of deploying police forces according to criminality.*OWA operator*: An OWA operator is a weighted average of the input values given an order induced among them [60,61]. The OWA operator has already been used to combine data using the Dempster-Shafer Theory, [59]. Given this situation, we introduce four operators: OWA-ASC OWA-DESC, weighted-OWA and Induced-OWA.*OWA-DESC operator*: When using this variant of the OWA operator values and weights are ordered both in descending order. This combination can emphasize the biggest belief values of each scenario, avoiding that a certain important fact known by one of the experts could be ignored because of the simple average of numbers. For example, if a crime scenario has a large belief degree at a certain location, then using average, this information can be mixed with lower degree values of the other scenarios.*OWA-ASC operator*: When using this variant of the OWA operator values are ordered in an ascending sequence and the weights are also ordered in an ascending sequence. This combination emphasizes the belief when the values are constantly high in all scenarios. This operator is similar to average, but it is not susceptible to isolated big values. It can be applied to allocate specific and limited resources that can support multiple scenarios. It can also be used to identify critical areas.*Weighted OWA operator*: The weighted OWA (WOWA) operator integrates the weighted average and the OWA operator in the same formulation. Thus, it can represent the importance of the variables and the attitudinal character of the decision maker in the same formulation, under or overestimating the initial data. The main advantage is that it can provide a more complete representation of the information taking into account any scenario that may occur between the minimum and the maximum. *Induced OWA operator*: The induced OWA (IOWA) operator is an aggregation operator that follows the methodology of the OWA operator [62]. However, instead of reordering in increasing or decreasing order, it uses order-inducing variables to determine the ordering process of the aggregation. This issue is important because many times the numerical values do no indicate the ordering of the information. The IOWA operator can be used to specify the scenario evaluation order. For example, if we want to order scenarios by their “source quality”, it is possible to define an order using the u values in IOWA pairs. However, the order cannot be arranged by an optimal value, because we are working with belief degrees. 

## 6. A Collaborative Decision-Making Supporting Tool Intended for Discussions on Preparedness for Emergencies

In this section, we present a tool that implements the collaborative decision model explained in the previous chapter by an example in which a team of three experts analyzes the tsunami scenario in Iquique. The members of the team have various expertise and/or information on identifying risk zones, people agglomerations and evacuation plans. We present screenshots of the most important stages of the tool with views of workspaces from the point of view of these three experts. The application obtains all the information needed in order to conduct this analysis from open public sources like OpenStreetMap (http://openstreetmap.org), public databases of Chilean ministries and National Office for Emergencies (ONEMI). 

The first view corresponds to the view a user has when logging into the platform. It shows the ongoing projects in evaluation/discussion, and the scenarios that participants recently generated which the user has not seen yet (Figure 4). It also shows the participants of the project, in this case three: Nelson, Jonathan and Alvaro. According to the screenshot, Nelson is the one who is logged in, Jonathan is also online but Alvaro is not. The screenshot shows there are three ongoing projects “Landslide prevention Chaiten”, “Evacuation planning Iquique”, and “Fire prevention Valparaiso”.

After selecting a project, the user can see the maps that have been generated for this project sorted by authorship (Figure 5) under the username of the team member who created them. 

By clicking the pushbutton labeled “new map” a user can generate a new map from scratch or by combining two or more already generated maps. First, we will explain the generation of a new map from scratch with an example. 

Let us assume the user Nelson has an expertise in analyzing urban areas where there can be agglomeration of people during a natural disaster. He has several hypotheses about why people concentrate in certain places and he wants to generate a map showing the number of people in the places near the coastal regions. Using the platform, he can include those hypotheses, giving them a percentage of mass and specifying exceptions that might occur (Figure 6). The platform takes this information and generates a belief map about which are the places where a large number of people could gather.

He thinks that there are usually high concentrations of people near amenity places, and near shops and at some hours of the day, near educational buildings. In order to evaluate his hypotheses separately, Nelson generates three scenarios (maps), one for each. 

On the other hand, Jonathan has expertise evaluating the level of risk that certain zones may have. He generates two risk scenarios, which evaluate the risk a person being in a certain geographical area of the city may face according to two parameters. The first one depicts the zones in which the risk of being flooded in case of a tsunami is determined according to their altitude (lower altitude means higher risk). In the second one, the risk depends on the distance that would be necessary to cover for a person to get out of the flood zones (longer distances means greater risk).

Álvaro is an expert in evacuation; he has only uploaded a map with the current evacuation routes and he must develop a new evacuation plan, which will use the scenarios created by Jonathan and Nelson. 

As a team, the three experts decide that in order to generate an evacuation plan, they require only one scenario showing the possible people concentration and another one that shows the risk zones, so both Nelson and Jonathan must combine their scenarios.

To combine the scenarios, the platform provides an interface to use the operators of addition, subtraction, average and ordered weighted averages (OWA). Nelson decides that the right operation for combining his maps should be the addition, since people in the concentration areas will sum up in a real scenario. Figure 7 shows a screenshot of the tool when combining the maps resulting from estimating the number of people in amenities and people at shops with the SUM operation, which is called “Sum People”. 

Jonathan decides that the most appropriate operation for combining the risk scenarios generated by him was a decreasingly ordered weighted average (OWA-DESC), which gives greater weighting to higher values and lower weighting to lower values. In this manner, the high-risk zones in any of the scenarios to be combined are maintained. The generated scenario is called “Danger DOWA” (Figure 8).

Finally, Alvaro combines the two scenarios generated by Jonathan and Nelson in a single one called “People in Danger”. In order to prioritize areas with high risk and at the same time areas with concentrations of people, he decides to combine using an OWA-ASC operator, being the areas that meet both conditions the most prominent (Figure 9).

The application also implements a tool that can be used for supporting the asynchronous discussion and the pertinence, validity or convenience of using a given scenario for preparing the actions needed to react in case of a disaster and/or plan the rescue and mitigation procedures. This tool consists of creating “Argumentation Objects”, e.g., to discuss the need to carry out a differentiated evaluation for night periods. In order to give context to the argument, it is possible to attach previously created scenarios and assign them a discussion category. Figure 10 shows a screenshot when user Nelson is creating an Argumentation Object.

The other project participants can review the argument with the attached antecedents (scenarios), discuss it and support the argument based on a voting system, which can be seen in Figure 11. 

## 7. Conclusions

The work reported here concerns the development of a model, a methodology and a computational system to support a collaborative decision-making process involving ill-structured problems where geographical data plays an important role. These are often called “suitability problems”. 

This research was motivated by the fact that Chile is a highly active seismic country and at the same time, it has a long coastland, with several cities along this coast. This exposes a large population to the possible scenario of a tsunami. A strong earthquake in 2010 which claimed hundreds of lives due to an afterwards tsunami and a moderate high earthquake in a coastal city of Iquique in 2014, set on the alarms at the need of developing evacuation plans for all inhabited coastal zones. However, although there are many systems supporting collaborative decision making in general and for particular situations, a literature review showed that there is no system intended for supporting the best evacuation routes decision process. In this work we have focused on studying some type of emergency preparation (planning sub-stage [63]), namely, preparations for tsunamis in a coastal city. As we have discussed, the problem also has epistemic uncertainty. However, preparing actions for other emergency situations could be also addressed using the results of this work. Other cases may be flood emergencies caused by rivers or by snow melting in mountains caused by sudden increases of temperature.

The presented approach uses Dempster-Shafer’s plausibility theory for analyzing emergency scenarios, since it allows to apply hypotheses and use uncertain data over geo-referenced information in order to draw conclusions about the necessary action that could be taken in case of a certain type of emergency. Then, we propose cooperative work aggregating the contributions of various experts, giving the possibility of discussing about the plausibility of the hypotheses stated by each expert participating in the preparedness team. The aggregation is done by combining suitability maps using operators previously proposed in the literature. The result is a few number of maps encapsulating the experts’ knowledge. Such maps may depict geographical areas, which are not appropriately supported to respond to the emergency. 

The proposal includes a computer tool (a CGIS) which supports experts to perform the two aforementioned stages. Antunes et al. [64] have presented several evaluation techniques for collaborative systems; the evaluation work described in this paper can be classified as a Knowledge-based scenario method. Another evaluation technique suitable for this kind of scenario tool could be a Scenario-Based Evaluation (SBE). The methods to validate scenarios like these ones require a long-term study, especially for evaluating tools used in emergency-related situations, since they are difficult to replicate in controlled environments for conducting evaluations. Therefore, we opted for validating the approach through a simulation of the preparation for a tsunami in a northern Chilean coastal city. Through this simulation, we show that the method and the developed tool indeed support the execution of Scenario-Based evaluation. Furthermore, we plan to conduct a user-centered evaluation method in the future to specifically evaluate usability with the tool; however, that is another inquiry project. 

In this work we could also check that the use of the Dempster-Shafer’s theory of plausibility is an appropriate approach for analyzing collaborative scenarios, since it allows multiple stakeholders to apply hypotheses and use uncertain data over geo-referenced information, in order to draw conclusions about the necessary action that could be taken in case of a certain type of problem. Furthermore, this work implements the aggregation of the contributions of various experts, giving the possibility of discussing about the plausibility of the hypotheses stated by each expert participating in the preparedness team.

## Figures and Tables

**Figure 1 sensors-19-05040-f001:**
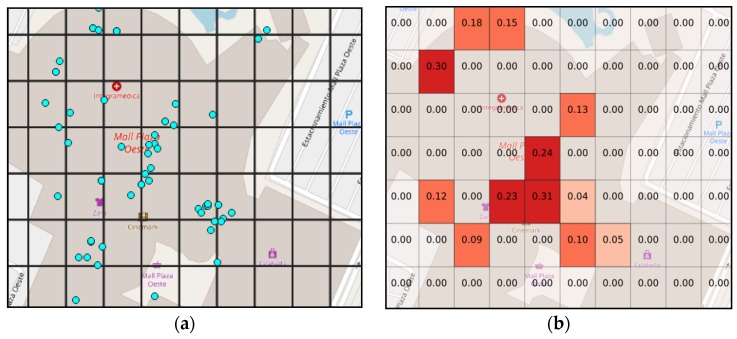
Example of geographical data (**a**) and belief computed for a certain region (**b**).

**Figure 2 sensors-19-05040-f002:**
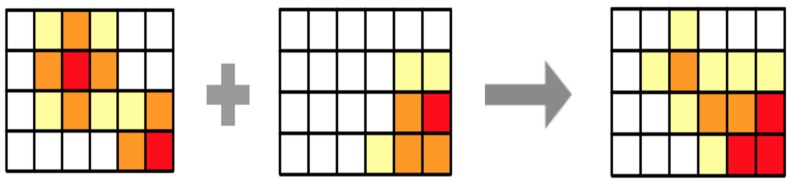
Example of geographical combination of belief maps from different hypotheses.

**Figure 3 sensors-19-05040-f003:**
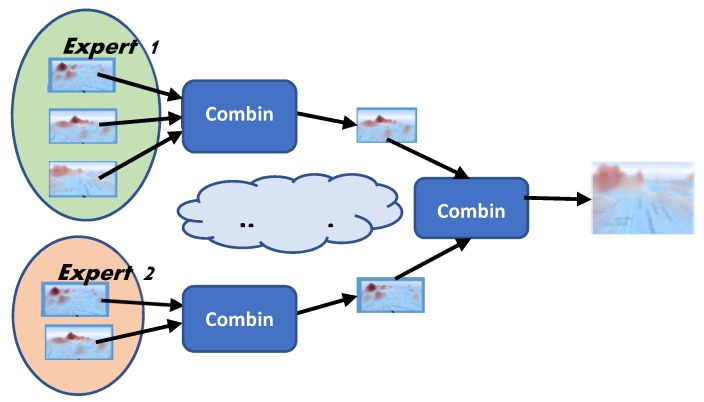
Hierarchical combination process. The figure shows that expert 1 generated three and expert 2 two scenarios individually. Expert 1 produced a single scenario combining the maps generated by her and presenting it to expert 2 for consideration. Expert 2 did the same with her two maps. After a discussion process (which is supported by the developed system) both experts agree to combine their scenarios in a single one, which is used for developing emergency preparedness plans.

**Figure 4 sensors-19-05040-f004:**
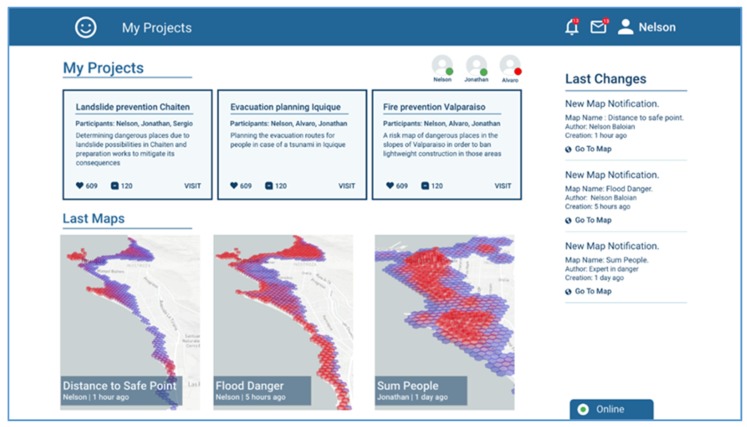
Screenshots of the application showing the available projects for user Nelson.

**Figure 5 sensors-19-05040-f005:**
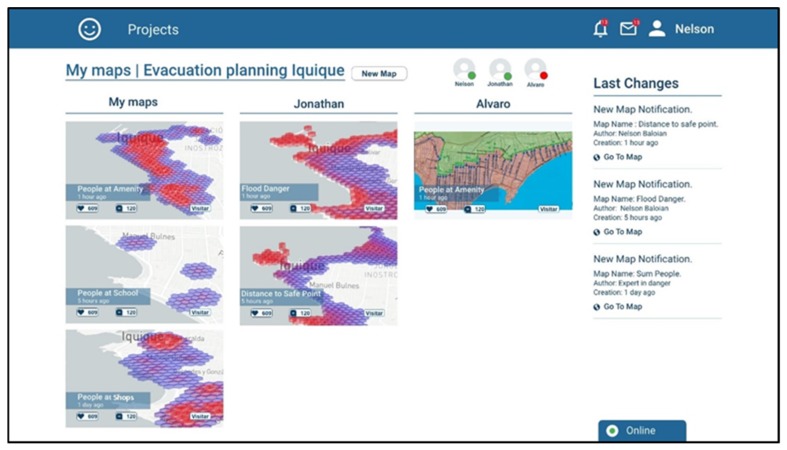
Screenshot showing the user Nelson exploring the maps that have been generated inside the “Evacuation planning Iquique” project.

**Figure 6 sensors-19-05040-f006:**
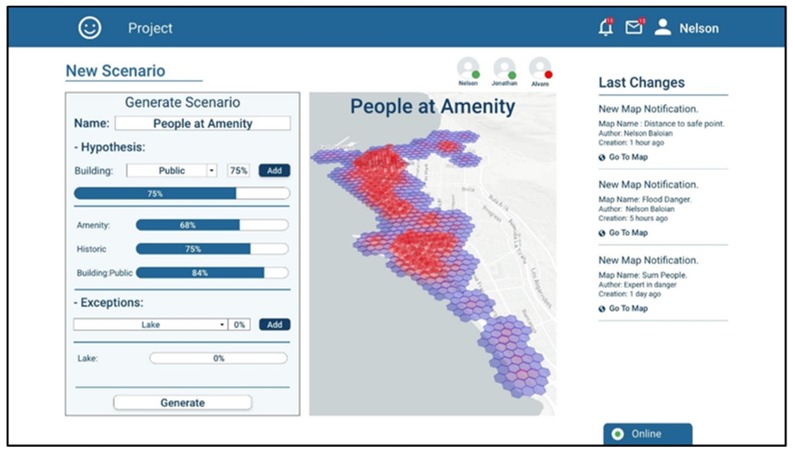
Screenshot of the application when generating a new map by defining hypotheses and their weights.

**Figure 7 sensors-19-05040-f007:**
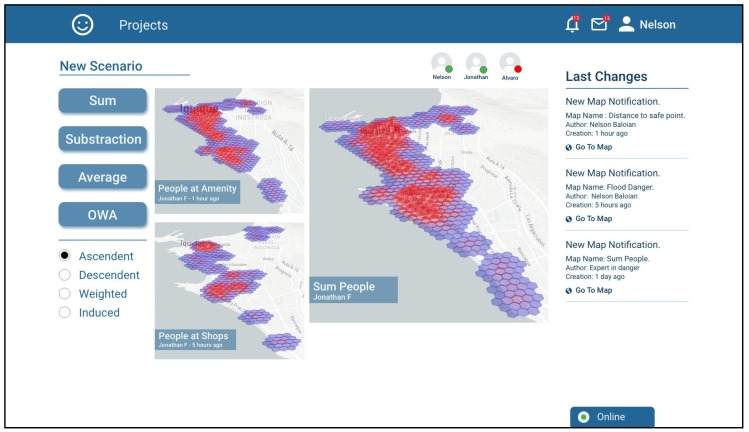
Screenshot of the application when Nelson combines the scenarios generated for people in amenities and people at shops with the sum operator.

**Figure 8 sensors-19-05040-f008:**
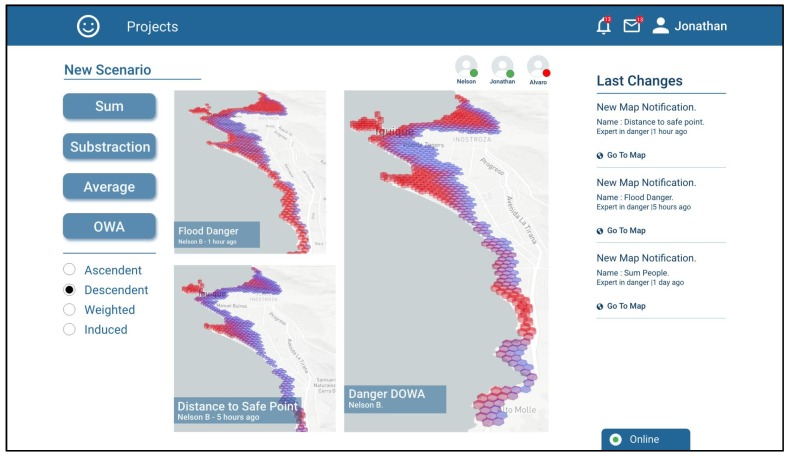
Screenshot showing Jonathan combining the two maps generated by him using the OWA-DESC operator.

**Figure 9 sensors-19-05040-f009:**
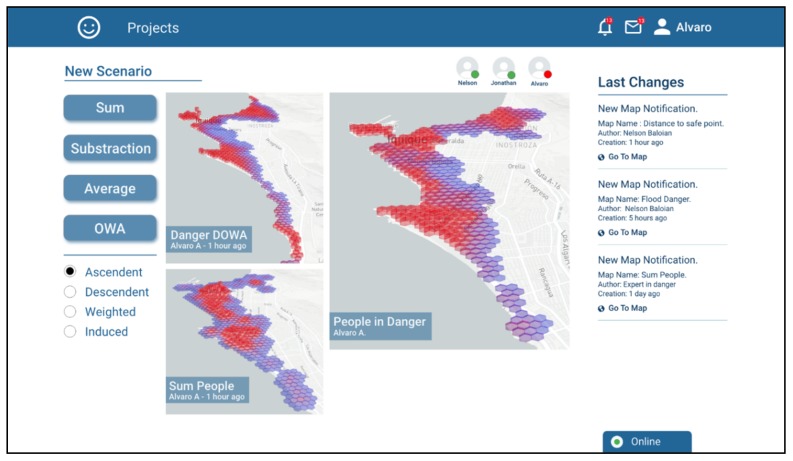
Screenshot showing Alvaro combining the two maps generated by him using the OWA-ASC operator.

**Figure 10 sensors-19-05040-f010:**
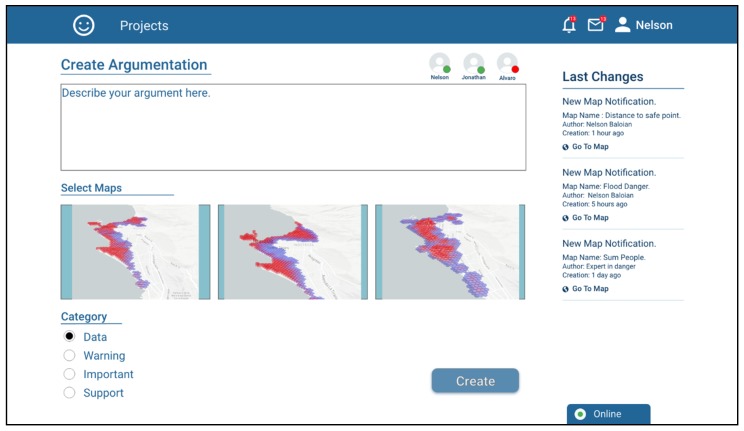
Creation of an Argumentation Object of the Data category.

**Figure 11 sensors-19-05040-f011:**
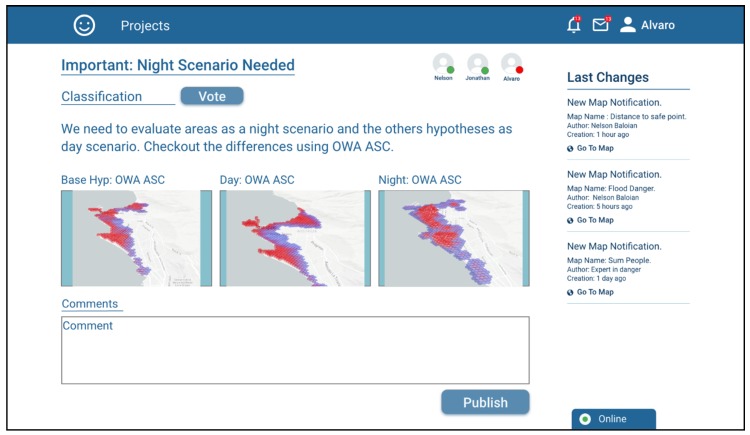
Screenshot of the voting and commenting function of the application.

**Table 1 sensors-19-05040-t001:** Description of most recent and relevant articles published on Emergency Preparedness.

Ref. No.	Contribution	Focus	Collaboration	Supports System Design
[39]	Explores the use of Gnutella peer-to-peer network over mobile ad-hoc networks in order to support large-scale CVEs	software architecture	yes	no
[40]	Reports on a study of one community’s emergency planning activities.	discussion	yes	no
[41]	Reviews the concepts of community preparedness and emergency planning	discussion guidelines	yes	no
[42]	Proposes national guidelines on disability inclusion in emergency preparedness	discussion	no	no
[43]	Emphasizes preparedness as an early stage for facing natural disasters	discussion	yes	no
[44]	Proposes a decision process for establishing an efficient network of secure storage	process proposal	no	no
[45]	Aimed to determine the degree to which Australia has worked on emergency preparedness for infant and young child feeding in emergencies	report	yes	no
[46]	Reports the levels of preparedness of a community exposed to two natural hazards and identifies the primary sociodemographic characteristics of groups with different preparedness levels	report/discussion	yes	no
[47]	Reviews the challenges and gaps of present disaster systems, establishing the root cause for failure as the lack of an effective mitigated disaster management system in place	process proposal	yes	no
[48]	Examines student preparedness perceptions, a better understanding of factors that may influence actual preparedness is needed.	report/discussion	no	no
[49]	Highlights the need for collaboration	discussion	yes	no
[50]	Presents the use of social media in emergency management.	process proposal	yes	no
[51]	Explores the appropriate planning for deployment of resources to provide relief to disaster victims and identifies which of these activities are critical to reduce suffering	process proposal	yes	no
[52]	Proposes a GIS Flowchart to determine the flood damage coefficient	process proposal	no	no
[53]	Proposes a novel mathematical model for redesigning existing relief logistics networks	model proposal	no	partially
[54]	Highlights the need for planning before the emergency occurs	discussion	no	no

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
