# Peer review of "Technology Support for Collaborative Preparation of Emergency Plans†"

_sensors, 2019, doi:10.3390/s19225040_

Round 1

Reviewer 1 Report

Emergency planning is an important early stage of disaster management, and the most challenge part in emergency planning is the scenario simulation or emergency prediction, which heavily depends on: relevant factors determination, near real-time data collection, and early-warning system development. The collaborative decision making system is important, but it should build on objective information (from either real-time sensing system or a prediction model), rather than subjective opinion. Although the Dempster-Shafer Theory mitigates such effect to some extent, this still cannot give review a strong confidence for emergency planning. GIS cannot be innovative by itself, it is only tool to support constructive ideas.

other minor mistakes:

please complete the sentence in line 95. Figure 4, caption should be next to the figure.

Author Response

Thank you for your evaluation. We have included sentences in the paper to incorporate your suggestions. We have added text to the introduction in order to clarify that we focus on natural disasters and the planning of actions and reactions before they happen, which is usually called the “preparedness stage” during which experts plan actions and measurements for responding in case of an emergency (for example, planning evacuation routes for the population in case of floods, fires, etc.). In particular, our country is heavily subject to earthquakes, which have been deadly (a Chilean region suffered the strongest earthquake in history: near Valdivia on May 22, 1960). Earthquakes, in turn, may generate tsunamis, and so, our example deals with this case. Unfortunately, earthquakes cannot be scientifically predicted, and so, real-time data is not useful for earthquake consequences planning. The suggestion of using real-time data is very important to be used for calling off a tsunami warning (and we have mentioned now that). Of course, authorities and people must know beforehand what to do in case of a strong earthquake: they should follow the prepared plan for coping with an eventual tsunami; if the authorities decide few minutes after an earthquake that the risk of a tsunami has decreased to null based on real-time data, then people can be advised to return to a normal situation. However, in case that the tsunami danger continues, then people should carry on the previously specified plans. Our proposal helps experts prepare such plans, based on objective data obtained from various sources. Here, your suggestion is again valuable: today, each expert can obtain her/his data from many sources, including simulation results.

Additionally, have improved the presentation by including additional references, especially those related to global innovative support for collaborative preparation of emergency plans. Particularly, we added a new section (2.3) in which we summarize our findings, identifying the generic aspects of the previous works on emergency preparedness and the research gap which our work aims to cover. For this purpose, we selected the most recent works.

Reviewer 2 Report

This manuscript contains exciting work from the technology support for collaborative preparation of emergency plans. However, the paper is written in style more like a lecture rather than a research article. The generic aspects and the global innovative technology support for collaborative preparation of emergency plans in research development haven't been presented. Some figures and tables which involve the world-wide novel research should be described and discussed with more details to emphasize the papers all over the world novelty. Please use the newest Web of Science journal papers.

Author Response

Thank you very much for your kind evaluation. We have improved the presentation by including additional references, especially those related to global innovative support for collaborative preparation of emergency plans. Particularly, we added a new section (2.3) in which we summarize our findings, identifying the generic aspects of the previous works on emergency preparedness and the research gap which our work aims to cover. For this purpose, we selected 16 of the most recent works.

Round 2

Reviewer 1 Report

All my previous concerns have been answered.

Author Response

Thank you for your review,

We added a couple of paragraphs at the conclusions section to highlight the novel contribution of this work to the field. 

Reviewer 2 Report

The manuscript worldwide ground-breaking novelty has not been emphasized.

Author Response

Thank you for your comments, We added a couple of paragraphs at the conclusions section to highlight the novel contribution of this work to the field.